# In Silico Analysis of Probiotic Bacteria Changes Across COVID-19 Severity Stages

**DOI:** 10.3390/microorganisms12112353

**Published:** 2024-11-18

**Authors:** Clarissa Reginato Taufer, Juliana da Silva, Pabulo Henrique Rampelotto

**Affiliations:** 1Graduate Program in Genetics and Molecular Biology, Universidade Federal do Rio Grande do Sul, Porto Alegre 91501-970, Brazil; clarissa.taufer@gmail.com (C.R.T.); juliana.silva@unilasalle.edu.br (J.d.S.); 2Graduate Program in Health and Human Development, Universidade La Salle, Canoas 92010-000, Brazil; 3Bioinformatics and Biostatistics Core Facility, Instituto de Ciências Básicas da Saúde, Universidade Federal do Rio Grande do Sul, Porto Alegre 91501-970, Brazil

**Keywords:** microbiome, gut microbiota, COVID-19, probiotics, biomarkers, disease severity, 16S rRNA, cross-cohort

## Abstract

The gut microbiota plays a crucial role in modulating the immune response during COVID-19, with several studies reporting significant alterations in specific bacterial genera, including *Akkermansia*, *Bacteroides*, *Bifidobacterium*, *Faecalibacterium*, *Lactobacillus*, *Oscillospira*, and *Ruminococcus*. These genera are symbionts of the gut microbiota and contribute to host health. However, comparing results across studies is challenging due to differences in analysis methods and reference databases. We screened 16S rRNA raw datasets available in public databases on COVID-19, focusing on the V3–V4 region of the bacterial genome. In total, seven studies were included. All samples underwent the same bioinformatics pipeline, evaluating the differential abundance of these seven bacterial genera at each level of severity. The reanalysis identified significant changes in differential abundance. *Bifidobacterium* emerged as a potential biomarker of disease severity and a therapeutic target. *Bacteroides* presented a complex pattern, possibly related to disease-associated inflammation or opportunistic pathogen growth. *Lactobacillus* showed significant changes in abundance across the COVID-19 stages. On the other hand, *Akkermansia* and *Faecalibacterium* did not show significant differences, while *Oscillospira* and *Ruminococcus* produced statistically significant results but with limited relevance to COVID-19 severity. Our findings reveal new insights into the differential abundance of key bacterial genera in COVID-19, particularly *Bifidobacterium* and *Bacteroides*.

## 1. Introduction

The COVID-19 pandemic remains a major global public health challenge. While the majority of infected individuals develop only mild to moderate symptoms, a substantial proportion of patients can progress to more severe clinical manifestations, including pneumonia, acute respiratory distress syndrome (ARDS), and multi-organ failure [1]. Several factors, including advanced age, underlying health conditions, and individual immune system characteristics, have been associated with the severity of COVID-19 [2,3]. The underlying processes driving the disease progression, nevertheless, are still mostly unclear.

Recently, the role of the gut microbiota in modulating the immune response and influencing COVID-19 progression has been increasingly investigated [4,5]. Studies have demonstrated that individuals with COVID-19 exhibit significant alterations in the composition of the microbiota, with a reduction or increase in the relative abundances of certain beneficial bacterial genera, such as *Akkermansia*, *Bacteroides*, *Bifidobacterium*, *Faecalibacterium*, *Lactobacillus*, *Oscillospira*, and *Ruminococcus* [6,7,8,9,10,11,12]. These bacterial genera are widely recognized for their immunomodulatory effects and protective properties against viral infections or for the protection and health of the host.

*Akkermansia* is an abundant microorganism in the human microbiota, capable of producing acetate and propionate from the degradation of mucin in the gut [13]. Additionally, this genus regulates the intestinal barrier, the expression of tight junction proteins, and is involved in immune homeostasis [13,14,15]. It also positively induces the formation of regulatory T cells (Tregs), contributing to the reduction in pro-inflammatory cytokines [16]. *Akkermansia muciniphila* has anti-influenza activity associated with its anti-inflammatory and immunoregulatory properties [17]. *Bacteroides* is one of the main genera of the gut microbiota, capable of modulating the intestinal barrier and involved in the degradation of polysaccharides and the production of short-chain fatty acids (SCFAs), which can enhance the antiviral response [18,19]. It also produces polysaccharide A, which enhances the immune response, protects against intestinal inflammation, and regulates natural resistance to viral infections [20,21,22]. *Bifidobacterium* species are among the most abundant members of the human gut microbiota. These bacteria have been extensively studied for their probiotic properties, including their ability to strengthen the gut mucosal barrier, modulate the immune system, and exert anti-viral effects [23,24]. *Bifidobacterium* can enhance the host’s defense against viral pathogens, such as influenza virus and rotavirus, by stimulating the production of antiviral cytokines and promoting the activation of natural killer cells [25,26]. *Faecalibacterium* is a butyrate-producing commensal bacterium, one of the most abundant in the gut, known for its anti-inflammatory properties and its role in protecting the intestinal barrier. In addition, *Faecalibacterium duncaniae* can alleviate the symptoms of flu caused by the H3N2 virus [27]. *Lactobacillus* species can modulate the host immune response through various mechanisms, including the production of SCFAs, the regulation of pro-inflammatory cytokines, and the enhancement of mucosal barrier integrity [28,29]. Several studies have demonstrated the beneficial effects of *Lactobacillus* in the prevention and management of viral respiratory infections, such as influenza and the common cold [30,31]. *Oscillospira* is frequently found in the gut microbiota and also has the capacity to produce SCFAs, primarily butyrate [32]. The abundance of *Oscillospira* was reduced in inflammatory diseases and depression-like behavior induced by lipopolysaccharide [33,34]. *Ruminococcus* is a prevalent member of the human core microbiota, with the capacity to degrade resistant starches and provide cross-feeding for other bacteria [35,36]. They express putative receptors for viruses and have been associated with negative effects in rotavirus infections in children and adults [37,38]. In addition, a strain of *Ruminococcus* has been shown to confer resistance to COVID-19, leading to the expansion of cytotoxic CD8+ T cells [39].

Therefore, changes in the abundance of these microorganisms may have implications for the severity of COVID-19. The depletion of these beneficial bacteria can potentially disrupt the balance of the host’s immune system, leading to an exaggerated inflammatory response and increased susceptibility to viral infections [40,41]. Furthermore, the metabolites produced by *Akkermansia*, *Bifidobacterium*, *Lactobacillus*, and *Faecalibacterium*, such as SCFAs and microbial anti-inflammatory molecule, possess anti-viral and anti-inflammatory properties that could be beneficial in the context of COVID-19 [42,43,44,45]. The disruption of the microbiome and the subsequent reduction in these protective metabolites may contribute to the dysregulation of the immune response and the development of more severe COVID-19 symptoms [46,47].

While many studies have explored the differences in the microbiome among COVID-19 patients, the direct comparison among them is challenging due to significant differences in data analysis procedures and the reference databases used for microbial identification and classification [48,49,50,51,52]. To address these limitations, the present study aims to perform a comprehensive reassessment of the microbiome datasets from studies that have investigated the gut microbiota in different stages of COVID-19. This approach allows for a standardized analysis and direct comparison of the relative abundances of key probiotic genera, providing a robust and comprehensive perspective on the changes in these beneficial microorganisms throughout the progression of the disease.

## 2. Materials and Methods

### 2.1. Study Selection and Obtaining the Sequencing Datasets

We screened the literature which analyzed the gut microbiome in COVID-19 by 16S rDNA gene sequencing and then selected studies that had their raw datasets available in public databases, such as the National Center Biotechnology Information (NCBI) and European Nucleotide Archive (ENA). We analyzed only stool 16S samples. We then selected studies that employed the V3–V4 hypervariable region. In addition, only studies that clearly defined the stages of COVID-19 were selected (i.e., mild, moderate, severe, and critical), ensuring no interventions were involved to avoid the impact of confounding factors. In total, 7 studies were selected for analysis (Table 1). Among them, one study had only a single defined group (PRJNA700830, critical), which excluded it from the individual analysis. However, this study was still included in the overall analysis.

### 2.2. Data Processing and Analysis of Sequencing Reads

The sequencing reads were processed using Mothur v.1.47.0 [53], adhering to a standard pipeline previously used in other studies [54,55,56]. The initial step involved removing barcodes and primers from the sequences (with no mismatches allowed), followed by a quality filter to eliminate low-quality reads. Quality control included trimming reads that exhibited low quality (Q < 30), incorrect lengths (minimum length = 270 bp, maximum length = 300 bp), ambiguous bases (maximum ambiguity = 0), or homopolymers longer than 6 bp. Potentially chimeric sequences were identified and discarded using VSEARCH [57]. Additionally, singletons were excluded to avoid incorporating sequences that might be artifacts resulting from PCR or sequencing errors [58]. After completing these initial quality filtering and trimming processes, the remaining sequences were clustered into Amplicon Sequence Variants (ASVs) and classified against the SILVA v.138 reference database [59]. Differences in the relative abundance of selected genera across various stages of COVID-19 were analyzed using LEfSe [60], with a corrected *p*-value of 0.05 considered statistically significant.

The analysis specifically concentrated on the V3–V4 regions, as it is well-established that the hypervariable regions of the 16S rRNA gene can significantly influence microbial composition. Focusing on a single region helps to reduce the biases that may occur from analyzing multiple 16S regions. Furthermore, most studies investigating gut microbiome changes associated with COVID-19 have utilized primers that target the V3–V4 hypervariable region.

## 3. Results

Table 1 summarizes all the information from the studies included in this reanalysis. In total, seven studies were selected, totaling 142 control samples and 439 samples from COVID-19 patients (first sample), consisting of 64 mild, 75 moderate, 182 severe, 27 critical, 91 COVID-19 samples without severity identification, 47 healthy controls, and 15 negative controls. The healthy and negative controls were analyzed together in the overall analysis (Figure 1, Appendix A). The negative controls refer to patients hospitalized for other reasons who had tested negative for COVID-19.

The severity classification used in the included studies was based on the “Clinical Management of COVID-19: Living Guideline” by the WHO [1] or the “Guidelines for the Diagnosis and Treatment of Coronavirus Disease 2019 (COVID-19) in China” [61]. Both definitions include mild, moderate, severe, and critical cases, with very similar classification criteria. Mild cases present mild clinical symptoms without signs of pneumonia, moderate cases show clinical signs of non-severe pneumonia, severe cases exhibit clinical signs of severe pneumonia, and critical cases involve ARDS symptoms, requiring mechanical ventilation and ICU admission.

The studies covered eight countries (Bangladesh, China, Germany, Italy, India, Switzerland, and Ireland), all from Europe and Asia, while no studies from America or Africa were included, highlighting a significant gap in the COVID-19-related mycobiome research globally. Regarding the NGS technology used, five studies employed the MiSeq sequencer, while two used NovaSeq 6000.

**Table 1 microorganisms-12-02353-t001:** Paper details identified from the studies included in the reanalysis.

Autor; Year	Accession Number	Country	Type of Study	NGS Technology	N	Groups
Albrich, W.C., et al.; 2022 [62]	PRJEB50040	Switzerland and Ireland	Cohort	MiSeq	98	8 mild, 24 moderate, and 66 severe
Gaibani, P., et al.; 2021 [63]	PRJNA700830	Italy	Case-control	MiSeq	69	COVID-19
Galperine, T., et al.; 2023 [64]	PRJEB61722	Switzerland	Cohort	MiSeq	57	42 severe, 15 critical
Rafiqul Islam, S.M., et al.; 2022 [65]	PRJNA767939	Bangladesh	Cross-section	MiSeq	37	15 healthy, 22 COVID-19
Reinol, J., et al.; 2021 [66]	PRJNA747262	Germany	Cross-section	NovaSeq 6000	212	95 negative, 44 mild, 35 moderate, 26 severe, 12 critical
Talukdar, D., et al.; 2023 [67]	PRJNA895415	India	Cohort	MiSeq	52	7 mild, 45 severe
Wu, Y. J., et al.; 2021 [68]	PRJNA684070	China	Case-control	NovaSeq 6000	56	32 healthy, 5 mild, 16 moderate, 3 severe

The PRJNA700830 study analyzed only COVID-19 patient samples, comparing them with healthy controls available in the databases. The PRJEB50040 study compared different severities of COVID-19, including severe survivors and fatal cases, as well as healthy controls. This study also used healthy controls selected from the databases. To reduce collection and sampling variations, we chose to use only the original controls from the studies.

The PRJNA767939 study compared COVID-19 patient samples without differentiating severities with original healthy controls. This was the only study reanalyzed for the comparison of healthy controls and COVID-19 (Table 2). The PRJEB61722 study analyzed samples from ventilated (ICU) and non-ventilated (non-ICU) patients. For comparisons, we classified ventilated patients as critical and non-ventilated patients as severe, following the characteristics of severity definition.

The PRJNA747262 study compared positive and negative patients and non-severe (mild + moderate) with severe (severe + critical) cases. Severity was reported for each sample, allowing for the evaluation of microbiota changes across all severities of COVID-19. The PRJNA895415 study compared mild and severe patients, and the PRJNA684070 study evaluated healthy controls (original), mild, moderate, and severe cases. These classifications were maintained in our reanalysis.

Table 3 shows the differential abundance results found using the Standard Protocol for the genera *Akkermansia*, *Bacteroides*, *Bifidobacterium*, *Faecalibacterium*, *Lactobacillus*, *Oscillospira*, and *Ruminococcus*, along with the original results from the studies. *Akkermansia* was not cited in the original studies, yet it was consistently identified in the analyses conducted using the standard protocol, which challenges the methodologies and implications of the findings. The analyses indicate that *Akkermansia* does not show significant abundance changes across the various stages of COVID-19 according to the standard protocol. While there are some variations in abundance levels, particularly in severe cases, the lack of consistent statistical significance suggests that *Akkermansia* may not be a key player in the disease’s progression.

*Bacteroides* was identified in all the studies through reanalysis but showed significant results only in the PRJNA747262 study. In the original studies, PRJEB50040 and PRJNA684070 did not report information on this genus in their results. In the PRJNA895415 study, the genus was enriched in patients with mild symptoms but was not statistically significant according to Lefse, as also observed in our reanalysis. Statistical significance in the original study was observed only in PRJNA747262 (*p* < 0.05 and LDA > 3.5), as noted in our reanalysis, which indicates a consistent increase in *Bacteroides* abundance from the healthy to severe stages.

*Bifidobacterium* was also identified in all studies through our reanalysis, showing statistical significance in four out of the five studies. Specifically, a significant decrease in the *Bifidobacterium* levels was observed in the critical and moderate stages of COVID-19. This decline raises questions about the potential implications of reduced *Bifidobacterium* in severe cases, suggesting that its depletion may be associated with adverse outcomes. The original results from the PRJNA895415 study demonstrated a significantly higher abundance of *Bifidobacterium* in patients with mild COVID-19, while PRJNA747262 identified it as a biomarker for COVID-19-negative patients. In the PRJEB50040 study, the genus showed a reduced relative abundance in high-risk patients. The findings from the standard protocol further corroborate these results, reinforcing the hypothesis that the presence of *Bifidobacterium* may be linked to disease severity. This suggests that higher levels of *Bifidobacterium* could potentially serve as a marker for milder forms of the disease, indicating its role in modulating immune responses during infection. The PRJNA684070 study compared COVID-19 patients with controls and identified an increase in *Bifidobacterium longum* in COVID-19 cases. This species is linked to increased ACE2 receptor expression in mice, which could impact viral contamination and disease progression [69]. The severity group analysis in our reanalysis aligns with findings from other studies, showing a reduction in *Bifidobacterium* as disease severity increases. The PRJEB61722 study did not provide information on this genus and was the only study that did not show significance in our reanalysis. These findings highlight the need for further investigation into the role of *Bifidobacterium* in COVID-19 progression and its potential as a therapeutic target or biomarker for disease severity.

*Faecalibacterium* was identified in all studies during reanalysis, except for PRJNA684070, which was also the only study that did not report information on this genus. However, none of our reanalyses showed statistically significant results. In the PRJNA747262 study, *Faecalibacterium* was identified as a discriminant between the patients with severe/critical COVID-19 (severe + critical) and those with non-severe disease (mild and moderate) in the original protocol, showing a reduction in severe/critical COVID-19 cases. In the PRJEB50040 study, the genus was associated with a lower-risk group. In our study, stratification by severity indicated that *Faecalibacterium* does not appear to be related to disease severity.

The abundance of *Lactobacillus* varied significantly across severity stages in different studies. The use of different reference databases and statistical methods appears to influence the detection and quantification of *Lactobacillus*. For the PRJNA684070 (*Lactobacillus* was elevated in the COVID-19 patients compared to the controls) and PRJNA895415 (enriched in the mild cases) studies, results were consistent between the original studies and the reanalysis. The PRJNA747262 study reported an increase in the Lactobacillales order in SARS-CoV-2-negative individuals compared to positive cases. PRJEB61722 and PRJEB50040 did not mention *Lactobacillus* in the text and showed no significance in our analyses.

*Oscillospira* was only observed in three studies during the reanalysis using a standard protocol (PRJEB61722, PRJNA747262, and PRJEB50040). In PRJEB61722 and PRJEB50040, the genus was statistically significant, but the LDA score was quite low, indicating that while *Oscillospira* may have shown some level of association with the studied conditions, its biological relevance might be limited. A low LDA score suggests that the effect size or strength of the association is not robust, which implies that *Oscillospira* does not play a major role in the microbiome’s changes related to the disease. In the original studies, this genus was not cited.

*Ruminococcus* was not identified in our analyses for the PRJNA684070 study, and it was also not cited in the original work. A statistical difference was observed only in PRJEB50040, which indicated a major increase in the abundance of *Ruminococcus* in mild cases compared to moderate and severe stages. In the original study, *Ruminococcus* was associated with lower-risk patients. *Ruminococcus* was observed in both the original study and the reanalysis in PRJNA895415, but it was not statistically significant. For PRJEB61722, the genus was identified as an increase in critical patients in a longitudinal assessment. Our analyses identified the presence of the genus, but the PRJNA747262 study did not cite it, and it was not statistically significant.

For the PRJNA767939 study, which compared samples from the COVID-19 patients to the healthy controls, the reanalysis is presented in Table 2*. Bifidobacterium* and *Bacteroides* exhibited significant increases in the COVID-19 patients across both analyses, with the standard protocol demonstrating more pronounced differences in their abundances. In contrast, *Akkermansia* and *Oscillospira* were not detected in either analysis, suggesting a potential absence of these genera in the studied populations. *Lactobacillus*, *Faecalibacterium*, and *Ruminococcus* were identified in the reanalysis, but they were not statistically significant. This discrepancy between the original study and the reanalysis in our study may be attributed to differences in the database used. While the NCBI database serves as a comprehensive repository, encompassing sequences from a wide array of organisms and samples, the Silva v. 138 database is an updated and curated collection specifically focused on bacterial sequences. This specialization enhances the alignment of sequences and improves the accuracy of microbial taxa identification, potentially leading to more reliable results in the reanalysis.

We also conducted an analysis of samples grouped by severity in the selected studies to evaluate the overall changes in these genera with relation to the severity of COVID-19 (Figure 1; Appendix A). For *Akkermansia*, no variation among the different levels of severity was observed (FDR = 0.12). As observed in the original projects, *Bacteroides* exhibited a distinctive pattern, with increased abundance in moderate cases and lower abundance in mild and severe cases, as well as critical cases differing from the healthy group. On the other hand, *Lactobacillus* presented an opposite profile when compared to *Bacteroides*, with decreased abundance in moderate cases and higher abundance in mild and severe cases, as well as critical cases not differing from the healthy group. The abundance of *Bifidobacterium*, *Faecalibacterium*, and *Ruminococcus* presented a statistically significant trend of reduction with increasing severity. The genus *Oscillospira* also presented a statistically significant trend of reduction with increasing severity (FDR = 0.004), but the LDAscore was quite low (0.1), indicating that the effect size was relatively small.

## 4. Discussion

In this work, we performed a standard bioinformatic protocol for the in silico analysis of publicly available microbiome datasets to further elucidate the dynamics of probiotic bacteria during the different stages of COVID-19. The reanalysis of microbial communities in COVID-19 patients revealed significant insights into the differential abundance of various genera, particularly *Bifidobacterium* and *Bacteroides*, while highlighting the limitations of previous studies. The discrepancies observed between the original studies and the reanalysis demonstrate the impact of methodological variations, including the choice of reference databases and statistical approaches, on the interpretation of microbial data.

Specifically, the choice of reference databases is critical; many original studies used databases that may not have provided comprehensive or accurate classifications of the microbial taxa [70,71,72]. This lack of representation can lead to the underreporting of certain genera, such as *Bifidobacterium*, which was consistently identified in our reanalysis using the more robust Silva v. 138 database.

Furthermore, the statistical approaches used in microbiome studies can greatly affect the interpretation of results. Inconsistent methodologies can yield varying conclusions about the abundance and significance of microbial genera, complicating the understanding of their roles in health and disease [73]. Our reanalysis, which applied a standardized protocol across datasets, revealed a more coherent picture of the microbial landscape in COVID-19 patients, emphasizing the importance of methodological rigor in drawing reliable conclusions.

By integrating data from multiple studies and employing a unified analytical framework, we were able to elucidate the complex of specific probiotic genera during COVID-19. In the original studies, *Akkermansia* was not detected, raising questions about its potential role in COVID-19. The reanalysis, however, consistently identified *Akkermansia* using the Silva v. 138 database, which is known for its comprehensive and updated taxonomic information. This finding suggests that the absence of *Akkermansia* in the original studies may be attributed to the use of less effective databases, such as Greengenes 13.8, which might not have adequately represented or classified this genus. Despite its detection in the reanalysis, the lack of significant abundance changes across disease stages implies that *Akkermansia* may not be a relevant factor in the progression of COVID-19 and cannot be used as a biomarker of the disease. Changes in *Akkermansia* in other viral infections appear related to the fast-spreading nature of the respective virus [74,75], indicating that changes in this taxon may be related to the consequences of viral infection on the microbiota, rather than influencing the protection or detriment of the host during the infection.

*Bacteroides* demonstrated a more complex pattern. While observed in both the original study and the reanalysis, statistical significance was only observed in the original and reanalysis for PRJNA747262. The steady increase in *Bacteroides* abundance from healthy to severe stages in PRJNA747262 suggests a potential association with disease severity. In this context, the increase in the opportunistic pathogen *Bacteroides nordii* was identified by Zuo et al. in COVID-19 patients [4]. Yeoh et al. found enriched *Bacteroides dorei* and *Bacteroides vulgatus* in COVID-19 patients, which are also involved in inflammatory bowel diseases [6]. Further investigation is needed to explore the role of *Bacteroides* in COVID-19 to understand whether its role is related to the inflammation associated with the disease or whether these bacteria are increased as opportunistic pathogens [6].

The profile observed for the genus *Lactobacillus* indicates a trend of increased abundance in more severe cases of the studied condition, contrasting with the decreased abundance of *Bacteroides* in similar situations. This divergence can be explained by the different ecological roles these genera play in the microbiome. *Lactobacillus* is often associated with beneficial effects on gut health, promoting intestinal barrier maintenance and modulating immune responses [76,77]. Its higher abundance in more severe cases may suggest an organismal attempt to restore intestinal homeostasis or protect against dysbiosis, even in the face of more severe clinical conditions. Additionally, this genus exhibits greater genetic diversity than what is typically observed and has recently undergone reclassification [78]. This has resulted in the creation of 23 new genera and the reassignment of some species that were previously classified under *Lactobacillus*. Although we used the most up-to-date database available, the nomenclature update for *Lactobacillus* has not yet been implemented. This suggests that the observations for *Lactobacillus* may not accurately reflect the current understanding of the genus.

In contrast, *Bacteroides*, which tends to be more abundant in moderate cases, may be involved in processes that promote inflammation or degrade components of the intestinal mucosa under stress [79,80]. Thus, the reduction in *Bacteroides* in more severe cases may reflect a microbiome response to pathological conditions, where an excess of this genus is not advantageous. These differences in abundance profiles may indicate adaptive mechanisms of the microbiome to various states of health and disease, suggesting that *Lactobacillus* may have a protective role, while *Bacteroides* may be more related to inflammatory or degradative processes.

*Bifidobacterium* emerged as a noteworthy genus in the analysis. The original studies indicated significant abundance in mild and moderate cases, a finding that was reinforced by the standardized reanalysis. This consistent observation supports the hypothesis that *Bifidobacterium* may be linked to milder disease forms, potentially influencing immune modulation during infection. In other COVID-19 studies, the species *Bifidobacterium bifidum* was negatively correlated with severity [6,81], and this species regulated the host’s innate immune response [82]. In mouse models, *Bifidobacterium pseudolongum* and *Bifidobacterium animalis* were elevated in the mice that survived an influenza infection, leading to the hypothesis that these species increase resistance to the virus through the modulation of the immune system and specific metabolic pathways related to the gut microbiome [83]. However, the significant decline in *Bifidobacterium* levels in critical and moderate stages observed in the reanalysis raises questions about its potential protective role and the implications of its depletion in severe cases. Other studies indicate a reduction in the species *Bifidobacterium adolescentis*, which also showed a trend of decreasing as disease severity increased [6,84,85]. These findings suggest that *Bifidobacterium* could serve as a valuable biomarker for disease severity and a potential therapeutic target.

The observed statistically significant reduction in the abundance of *Faecalibacterium* and *Ruminococcus* with the increasing severity of the condition suggests a potential disruption in the gut microbiota associated with disease progression. Both genera are known for their beneficial roles in maintaining gut health and homeostasis. *Faecalibacterium prausnitzii*, in particular, is recognized for its anti-inflammatory properties and its capacity to produce SCFAs, which are essential for intestinal health and the regulation of immune responses [86]. A decline in *Faecalibacterium* abundance may indicate a loss of these protective effects, potentially contributing to increased intestinal inflammation and a compromised mucosal barrier. Similarly, the *Ruminococcus* species are involved in the fermentation of dietary fibers, playing a major role in SCFA production and overall gut metabolism [87,88]. Their reduction could lead to a decrease in metabolic functions within the gut, exacerbating dysbiosis and potentially influencing the severity of the clinical condition. The correlation between the decrease in these beneficial taxa and the increase in disease severity highlights the importance of microbial diversity and composition in maintaining health. This trend may reflect an underlying mechanism where the depletion of beneficial microbes contributes to a pathological state, reinforcing the notion that microbial dysbiosis can play a critical role in the progression of various diseases. Further studies into the functional implications of reduced *Faecalibacterium* and *Ruminococcus* could provide deeper insight into their roles in COVID-19, as well as potential therapeutic opportunities aimed at restoring the microbial balance in affected individuals.

*Oscillospira*’s presence in the reanalysis, albeit with low statistical significance and LDA scores, indicates that while there is a notable decrease in the abundance of *Oscillospira* with increasing severity, the effect size is relatively small. This suggests that although *Oscillospira* may play a role in the microbial community associated with COVID-19, its contribution might not be as impactful as the other taxa with higher LDA scores. Further investigation is needed to understand the ecological significance of *Oscillospira* in this context and whether its reduction is a consequence of the disease or a contributing factor to its progression. In addition, exploring interactions with other microbial taxa through network analysis may provide insights into the complex dynamics of the microbiome in relation to disease severity. In any case, the relatively small effect size highlights the necessity for cautious interpretation of microbial associations, as low effect sizes can undermine the robustness of conclusions drawn from such data.

The findings from this study establish a strong basis for identifying microbiome biomarkers linked to COVID-19 and highlight their potential application as probiotics for adjuvant treatment of the disease. The significant differential abundance of genera such as *Bifidobacterium*, *Bacteroides*, and *Faecalibacterium* across the various stages of COVID-19 suggests that these taxa may serve as indicators of disease severity. Particularly, the consistent presence of *Bifidobacterium* in milder cases could indicate a protective role, potentially influencing immune modulation during infection. This opens up promising new opportunities for developing probiotics that enhance the abundance of beneficial microbes, thereby supporting the immune response and potentially mitigating the severity of COVID-19. Furthermore, the identification of specific microbial patterns associated with disease progression could lead to the establishment of microbiome-based biomarkers that aid in patient stratification and personalized treatment approaches. With the integration of microbiome analysis into clinical practice, it may be possible to develop adjuvant therapies that leverage the gut microbiota to improve patient outcomes in COVID-19.

In addition, this study presents several significant strengths that enhance our understanding of the microbiome’s role in COVID-19. The application of a robust bioinformatic protocol for the in silico analysis of publicly available microbiome datasets allows for a consistent interpretation of microbial data, while the comprehensive reanalysis of multiple datasets provides a broader perspective on the dynamics of probiotic bacteria during different stages of the disease. By emphasizing methodological rigor, particularly through the use of a more comprehensive reference database (Silva v. 138), our study effectively addresses previous discrepancies and the underreporting of key genera, such as *Akkermansia*, *Bifidobacterium*, and *Lactobacillus*.

However, this study is not without its limitations. The reliance on publicly available microbiome datasets may introduce variability due to differences in sample collection, processing, and sequencing methodologies across studies. Additionally, while the reanalysis provided a more coherent picture of microbial dynamics, it is still constrained by the inherent limitations of the original datasets. Future studies should focus on longitudinal analyses with standardized protocols to better understand the temporal dynamics of the microbiome in COVID-19 patients. Furthermore, experimental studies are needed to elucidate the functional roles of the identified microbial taxa and their interactions with host immune responses. This could contribute to targeted probiotic interventions and further validate the potential of microbiome biomarkers in clinical settings.

## 5. Conclusions

This study provides a comprehensive reanalysis of publicly available microbiome datasets to investigate the dynamics of beneficial bacterial genera during different stages of COVID-19. Our findings reveal significant insights into the differential abundance of key genera, particularly *Bacteroides*, *Bifidobacterium*, and *Faecalibacterium*, emphasizing their potential roles in modulating the immune response and influencing disease severity. The differences observed between the original studies and our reanalysis underscore the critical impact of methodological variations, such as the choice of reference databases and statistical approaches, on the interpretation of microbial data. Using a standardized analytical framework allowed us to achieve more reliable and consistent results, refining our understanding of the microbiome’s role in COVID-19. In addition, these insights also highlight the need for rigorous methodologies in microbiome research. Future studies should continue to explore the therapeutic potential of beneficial bacteria in the context of COVID-19 and other diseases. Ultimately, this work provides the basis for further studies that may lead to innovative strategies for managing infections through microbiome modulation, paving the way for improved health outcomes.

## Figures and Tables

**Figure 1 microorganisms-12-02353-f001:**
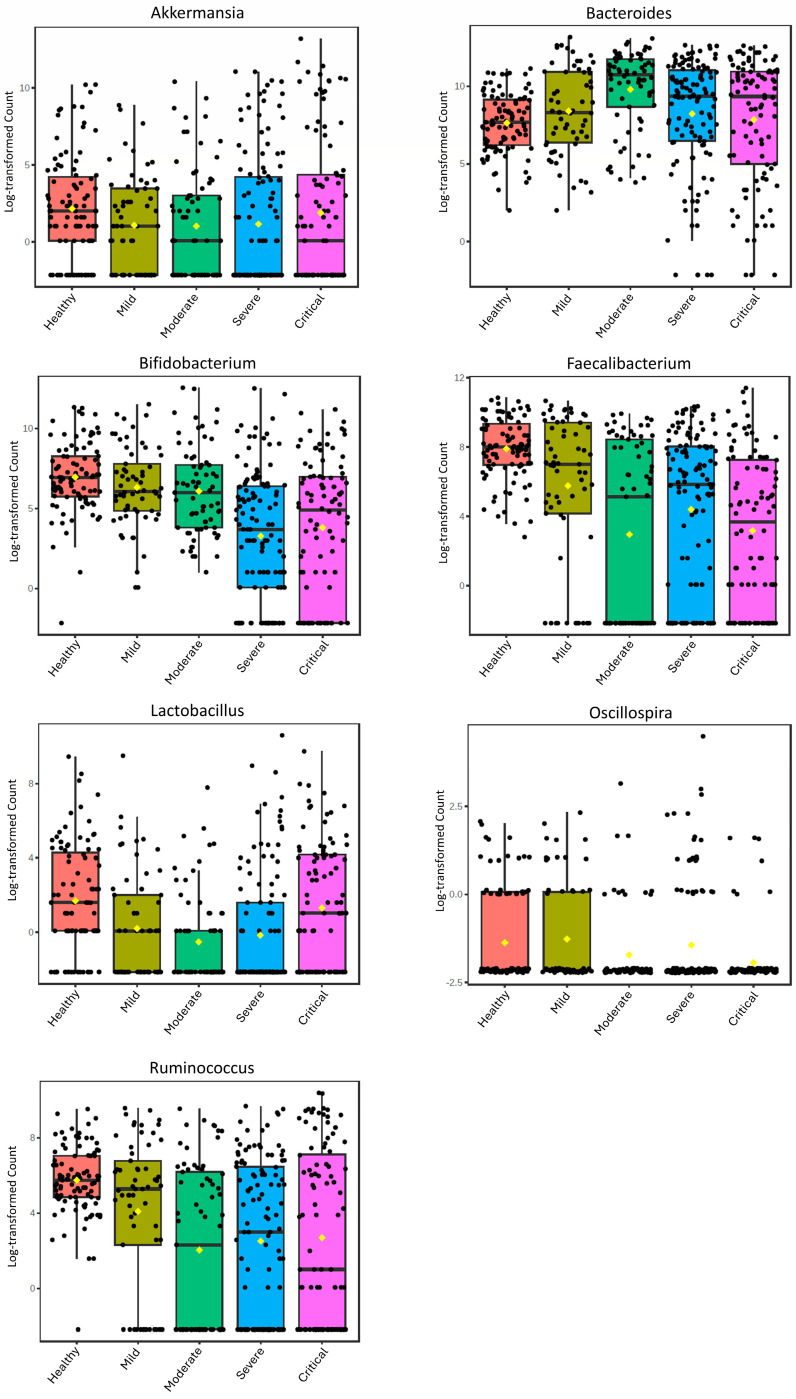
Boxplots of the relative abundance of the genera *Akkermansia*, *Bacteroides*, *Bifidobacterium*, *Faecalibacterium*, *Lactobacillus*, *Oscillospira*, and *Ruminococcus* in healthy controls and at different levels of severity of COVID-19. The Lefse analysis is presented in Appendix A.

**Table 2 microorganisms-12-02353-t002:** Differential abundance of the genera in healthy controls vs. COVID-19 patients according to the original study and the re-analysis using a standard protocol. Only one study was eligible for this analysis. Bold numbers indicate statistical significance.

Genus	Study	Reference Database	Statistical Analysis	*p* Values	FDR	Healthy	COVID-19	LDA Score
*Akkermansia*	PRJNA767939	NCBI	Kruskal-Wallis	-	-	-	-	-
	Standard Protocol	Silva v. 138	LEfSe	-	-	-	-	-
*Bacteroides*	PRJNA767939	NCBI	Kruskal-Wallis	**0.0039**	-	2.7 ± 0.6	5.6 ± 0.6	-
	Standard Protocol	Silva v. 138	LEfSe	**0.00546**	**0.039524**	43.933	692.86	2.51
*Bifidobacterium*	PRJNA767939	NCBI	Kruskal-Wallis	0.0036	-	2.9 ± 0.4	4.8 ± 0.4	-
	Standard Protocol	Silva v. 138	LEfSe	**0.00811**	0.490316	21.133	337.23	2.2
*Faecalibacterium*	PRJNA767939	NCBI	Kruskal-Wallis	-	-	-	-	-
	Standard Protocol	Silva v. 138	LEfSe	0.01214	0.0601	167.93	167.91	0.005
*Lactobacillus*	PRJNA767939	NCBI	Kruskal-Wallis	-	-	-	-	-
	Standard Protocol	Silva v. 138	LEfSe	0.22587	0.43817	72.667	0	0.666
*Ruminococcus*	PRJNA767939	NCBI	Kruskal-Wallis	-	-	-	-	-
	Standard Protocol	Silva v. 138	LEfSe	0.73108	0.89347	0.53333	16.818	−0.197
*Oscillopira*	PRJNA767939	NCBI	Kruskal-Wallis	-	-	-	-	-
	Standard Protocol	Silva v. 138	LEfSe	-	-	-	-	-

**Table 3 microorganisms-12-02353-t003:** Differential abundance of *Akkermansia*, *Bacteroides*, *Bifidobacterium*, *Faecalibacterium*, *Lactobacillus*, *Oscillospira*, and *Ruminococcus* in each disease stage according to the original study and the re-analysis using a standard protocol. Bold numbers indicate statistical significance. “-” indicates that the genus was not cited; “NR” indicates values not reported.

Genus	Study	Reference Database	Statistical Analysis	*p* Values	FDR	Healthy	Mild	Moderate	Severe	Critical	LDA Score
*Akkermansia*	PRJEB50040	NR	Mann–Whitney	-	-	-	-	-	-	-	-
	Standard Protocol	Silva v. 138	LEfSe	0.20718	0.64559	-	118.33	219.32	23.375	-	2
	PRJEB61722	EzBioCloud	NBZIMM	-	-	-	-	-	-	-	-
	Standard Protocol	Silva v. 138	LEfSe	0.25462	0.71539	-	-	-	339.76	349.87	0.782
	PRJNA747262	Greengenes 13.8	LEfSe	-	-	-	-	-	-	-	-
	Standard Protocol	Silva v. 138	LEfSe	0.26107	0.62404	209.45	100.77	258	762.92	81.667	2.53
	PRJNA895415	Silva v. 138	LEfSe	-	-	-	-	-	-	-	-
	Standard Protocol	Silva v. 138	LEfSe	0.61856	0.80146	-	18.429	-	99	-	1.62
	PRJNA684070	Greengenes 13.8	LEfSe	-	-	-	-	-	-	-	-
	Standard Protocol	Silva v. 138	LEfSe	0.14288	0.31141	-	0.1	48	0	-	1.4
*Bacteroides*	PRJEB50040	NR	Mann–Whitney	-	-	-	-	-	-	-	-
	Standard Protocol	Silva v. 138	LEfSe	0.85223	0.93806		615.62	708.12	521.25		1.98
	PRJEB61722	EzBioCloud	NBZIMM	-	-	-	-	-	-	-	-
	Standard Protocol	Silva v. 138	LEfSe	0.31891	0.71539	-	-	-	6377.7	4934.2	2.86
	PRJNA747262	Greengenes 13.8	LEfSe	**<0.05**	NR	NR	NR	NR	NR	NR	>3.5
	Standard Protocol	Silva v. 138	LEfSe	**0.015893**	0.15148	1294.7	2000.2	2796.6	2225.7	3793.8	3.1
	PRJNA895415	Silva v. 138	LEfSe	0.47709	0.67512	-	NR	-	NR	-	5.03
	Standard Protocol	Silva v. 138	LEfSe	0.9893	0.9893	-	1156.6	-	1099.2	-	1.47
	PRJNA684070	Greengenes 13.8	LEfSe	-	-	-	-	-	-	-	-
	Standard Protocol	Silva v. 138	LEfSe	0.19739	0.38132	-	10796	7530.5	7351.2	-	3.24
*Bifidobacterium*	PRJEB50040	NR	Mann–Whitney	NR	NR	-	NR	NR	NR	-	NR
	Standard Protocol	Silva v. 138	LEfSe	**0.014983**	0.20259	-	175.04	88.652	158.62	-	1.65
	PRJEB61722	EzBioCloud	NBZIMM	-	-	-	-	-	-	-	-
	Standard Protocol	Silva v. 138	LEfSe	0.39997	0.71539	-	-	-	92.476	27.733	1.52
	PRJNA747262	Greengenes 13.8	LEfSe	**<0.05**	NR	NR	NR	NR	NR	NR	>3.5
	Standard Protocol	Silva v. 138	LEfSe	**0.036361**	0.25788	1014.1	673.86	728.63	1067.8	240.42	2.62
	PRJNA895415	Silva v. 138	LEfSe	**0.0014225**	**0.027866**	-	NR	-	NR	-	5.69
	Standard Protocol	Silva v. 138	LEfSe	**0.0091114**	0.21696	-	1631.3	-	574.96	-	2.72
	PRJNA684070	Greengenes 13.8	LEfSe	**0.000000788**	-	-	NR	NR	NR	-	2.976288515
	Standard Protocol	Silva v. 138	LEfSe	**0.00093705**	**0.022593**	-	628.1	1315.9	28.333	-	2.82
*Faecalibacterium*	PRJEB50040	NR	Mann–Whitney	NR	NR	-	NR	NR	NR	-	NR
	Standard Protocol	Silva v. 138	LEfSe	0.10236	0.51668	-	192.88	223.17	441.62	-	2.1
	PRJEB61722	EzBioCloud	NBZIMM	NR	NR	-	-	-	NR	NR	NR
	Standard Protocol	Silva v. 138	LEfSe	0.11045	0.65431	-	-	-	362.33	711.02	2.24
	PRJNA747262	Greengenes 13.8	LEfSe	**<0.05**	NR	NR	NR	NR	NR	NR	>3.5
	Standard Protocol	Silva v. 138	LEfSe	0.13339	0.49392	1350	1356.6	1123.1	859.73	544.33	2.61
	PRJNA895415	Silva v. 138	LEfSe	0.33006	0.55638	-	NR	-	NR	-	4.94
	Standard Protocol	Silva v. 138	LEfSe	0.19233	0.7538	-	549.57	-	244.44	-	2.19
	PRJNA684070	Greengenes 13.8	LEfSe	-	-	-	-	-	-	-	-
	Standard Protocol	Silva v. 138	LEfSe	-	-	-	-	-	-	-	-
*Lactobacillus*	PRJEB50040	NR	Mann–Whitney	-	-	-	-	-	-	-	-
	Standard Protocol	Silva v. 138	LEfSe	0.28178	0.68442	-	0.41667	18.182	0	-	0.281
	PRJEB61722	EzBioCloud	NBZIMM	-	-	-	-	-	-	-	-
	Standard Protocol	Silva v. 138	LEfSe	0.60958	0.81735	-	-	-	18.738	13.267	0.572
	PRJNA747262	Greengenes 13.8	LEfSe	-	-	-	-	-	-	-	-
	Standard Protocol	Silva v. 138	LEfSe	**0.036361**	0.25788	1014.1	673.86	728.63	1067.8	240.42	2.62
	PRJNA895415	Silva v. 138	LEfSe	**0.012934**	**0.058699**	-	NR	-	NR	-	4.73
	Standard Protocol	Silva v. 138	LEfSe	**0.0091114**	0.21696	-	1631.3	-	574.96	-	2.72
	PRJNA684070	Greengenes 13.8	LEfSe	**0.017048742**	-	-	NR	NR	NR	-	2.831048869
	Standard Protocol	Silva v. 138	LEfSe	**0.0043056**	**0.038006**	-	0.4	0.35294	20.833	-	1.05
*Oscillospira*	PRJEB50040	NR	Mann–Whitney	-	-	-	-	-	-	-	-
	Standard Protocol	Silva v. 138	LEfSe	**0.00044315**	**0.033128**	-	0.625	0.12121	0.375	-	0.1
	PRJEB61722	EzBioCloud	NBZIMM	-	-	-	-	-	-	-	-
	Standard Protocol	Silva v. 138	LEfSe	**0.014872**	0.47234	-	-	-	0.53333	34.762	0.393
	PRJNA747262	Greengenes 13.8	LEfSe	-	-	-	-	-	-	-	-
	Standard Protocol	Silva v. 138	LEfSe	0.78601	0.8749	14.842	17.955	21.429	22.692	10.833	0.202
	PRJNA895415	Silva v. 138	LEfSe	-	-	-	-	-	-	-	-
	Standard Protocol	Silva v. 138	LEfSe	-	-	-	-	-	-	-	-
	PRJNA684070	Greengenes 13.8	LEfSe	-	-	-	-	-	-	-	-
	Standard Protocol	Silva v. 138	LEfSe	-	-	-	-	-	-	-	-
*Ruminococcus*	PRJEB50040	NR	Mann–Whitney	NR	NR	-	NR	NR	NR	-	NR
	Standard Protocol	Silva v. 138	LEfSe	**0.0011837**	0.05019	-	197.71	52.182	93.875	-	1.87
	PRJEB61722	EzBioCloud	NBZIMM	NR	NR	-	-	-	NR	NR	NR
	Standard Protocol	Silva v. 138	LEfSe	0.060367	0.58486	-	-	-	71	191.17	1.79
	PRJNA747262	Greengenes 13.8	LEfSe	-	-	-	-	-	-	-	-
	Standard Protocol	Silva v. 138	LEfSe	0.29349	0.62404	318.84	508.55	439.37	560.35	678.17	2.26
	PRJNA895415	Silva v. 138	LEfSe	0.60229	0.78007	-	NR	-	NR	-	3.29
	Standard Protocol	Silva v. 138	LEfSe	0.48934	0.76757	-	24.571	-	24.511	-	0.0129
	PRJNA684070	Greengenes 13.8	LEfSe	-	-	-	-	-	-	-	-
	Standard Protocol	Silva v. 138	LEfSe	-	-	-	-	-	-	-	-

## Data Availability

The original contributions presented in the study are included in the article/Appendix A, further inquiries can be directed to the corresponding author.

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
