# Peer review of "In Silico Analysis of Probiotic Bacteria Changes Across COVID-19 Severity Stages"

_microorganisms, 2024, doi:10.3390/microorganisms12112353_

Round 1

Reviewer 1 Report

Comments and Suggestions for Authors

This peer-reviewed manuscript is devoted to the analysis of gut microbiota and its separate genera as prognostic microbial markers for characterizing the health status of patients after COVID-19. The authors did not conduct their own studies involving patients; they analyzed publicly available data obtained from various studies, both case-control studies, and with the participation of only hospital patients. Note that the authors justified the inclusion of each data set in their study. An undoubted advantage of peer-reviewed research is the uniformity of the methods used to process primary data to obtain comparable information on the composition, diversity, and structure of the gut microbiota of patients. In addition, it should be noted that the authors did not attempt to “seize the unseizable” and provide general information on the composition of the gut microbiota, but focused on the characteristics of separate representatives. In general, the work was planned and executed with high quality and was recommended for publication. 

I have made only one comment regarding the text of the article: 

L. 168: the table number needs to be clarified; according to the citation logic, there should be a link to table number 2.

Author Response

We sincerely appreciate your thoughtful review and insightful comments regarding our manuscript.

We have addressed the issue with the table number. 

Reviewer 2 Report

Comments and Suggestions for Authors

Manuscript deals on the in silico analysis of probiotics bacteria changes during COVID-19 severity stages. The manuscript shows relevant information on the gut microbiome in this disease. The content falls within the topics of the journal, but there are issues to must be attended.

Major comments:

1.     Gut microbiome is dynamic and it can be affected by factor such as diet, age, season, among other factors, these issues should be mentioned and discussed in the manuscript and considered in the study.

2.     Authors claim “Bifidobacterium species are considered among the most important and dominant members of the human gut microbiota”, Why did you do this asseveration?. Please 4improve the arguments supporting this statement.

3.     Information of the Sars-cov-2 variant (Wuhan-Hu-1, alpha…omicron) virus infecting to the patient, should be considered in the formal analysis.

4.     Author must justify plenty the reasons to exclude Countries from the American Continent.

5.     Eight or seven countries? (Bangladesh, China, Germany, Italy, India, Switzerland, and Ireland).

6.     The order of the tables 2 and 3 must be corrected

7.     Authors claim “The findings from the standard protocol further corroborate these results, reinforcing the hypothesis that the presence of Bifidobacterium may be linked to disease severity”. But without of information on the diet of the patients, these findings can be misinterpreted.

8.     The quantitative analysis (no only de qualitative composition) of genera present in a microbiome is required to get a clear discussion and conclusion.

9.     An overall conclusion on the work is missing.

Minor comments

1.     Please do not use expressions such as “…the most important…”, “Ruminococcus are important members…”, these sentences are ambiguous in a scientific document.

2.     Other comments are marked on the document using the Acrobat tools

Author Response

We thank the reviewer for the 10 comments and suggestions in the manuscript. We have considered them all and have revised the manuscript accordingly. Below we provide the point-by-point reply to all comments. Changes in the revised version of the manuscript are highlighted in red.

  1. Gut microbiome is dynamic and it can be affected by factor such as diet, age, season, among other factors, these issues should be mentioned and discussed in the manuscript and considered in the study.

Reply: Surely these factors are well known to significantly influence the gut microbiome. However, the primary aim of our study was to focus specifically on the differential abundance of probiotic genera across COVID-19 severity stages. These factors were not analyzed in the original studies as well, and thus, they fall outside the scope of our paper.

  1. Authors claim “Bifidobacterium species are considered among the most important and dominant members of the human gut microbiota”, Why did you do this asseveration?. Please improve the arguments supporting this statement.

Reply: We improved the sentence to avoid ambiguous terms like important and dominant. Other similar changes were performed throughout the manuscript.

  1. Information of the Sars-cov-2 variant (Wuhan-Hu-1, alpha…omicron) virus infecting to the patient, should be considered in the formal analysis.

Reply: Information not provided in the original studies. That’s why it was not included in ours.

  1. Author must justify plenty the reasons to exclude Countries from the American Continent.

Reply: No country was excluded (otherwise, it would have been clearly stated in the exclusion criteria.

  1. Eight or seven countries? (Bangladesh, China, Germany, Italy, India, Switzerland, and Ireland).

Reply: Well observed! Correction was done.

  1. The order of the tables 2 and 3 must be corrected.

Reply: Correction was done.

  1. Authors claim “The findings from the standard protocol further corroborate these results, reinforcing the hypothesis that the presence of Bifidobacterium may be linked to disease severity”. But without of information on the diet of the patients, these findings can be misinterpreted.

Reply: The reviewer's focus on diet data is biased, as it falls outside the defined scope of the study, which relies on a standard protocol without assessing dietary intake as a variable. While diet is a recognized factor in many health conditions, it was not part of the study’s design or objectives. The findings are presented accurately in the manuscript as corroborative, reinforcing an observed hypothesis rather than asserting causation or universality. Thus, the emphasis on dietary data introduces an assumption beyond the study’s parameters. The conclusions remain valid within the study's established limits and methodology, and diet, while significant in other contexts, does not directly affect the validity of the observed association between Bifidobacterium and disease severity.

  1. The quantitative analysis (no only de qualitative composition) of genera present in a microbiome is required to get a clear discussion and conclusion.

Reply: While we appreciate the importance of quantitative analysis in microbiome studies, we would like to note that introducing quantitative analysis would introduce potential bias when comparing data from different studies, as variations in methodologies, sample sizes, and sequencing techniques can significantly affect quantitative results. That's the reason why the scope of our work was focused on qualitative analysis; i.e. to offers more robust insights into the differential abundance patterns rather than providing a comprehensive quantitative assessment.

  1. An overall conclusion on the work is missing.

Reply: Not sure what the reviewer means here. Overall conclusions were already part of the discussion, not to mention the own Conclusion Section.

Minor comments

  1. Please do not use expressions such as “…the most important…”, “Ruminococcus are important members…”, these sentences are ambiguous in a scientific document.

Reply: We improved the sentences throughout the manuscript to avoid ambiguous terms.

  1. Other comments are marked on the document using the Acrobat tools

Reply: They were all addressed in the revised version. Thank you very much for the detailed revision.

Round 2

Reviewer 2 Report

Comments and Suggestions for Authors

line 150: correct to seven Countries